# Seroprevalence of antibodies against SARS-Cov-2 in the high impacted sub-district in Jakarta, Indonesia

Olivia Herlinda[1]☯, Adrianna Bella[1]☯*, Gita Kusnadi[1], Dimitri Swasthika Nurshadrina[1], Mochamad Thoriq Akbar[1], Sofwatun Nida[1], Ngabila Salama[2], Iwan Ariawan[3], Diah Saminarsih[4]

1 Center for Indonesia's Strategic Development Initiatives (CISDI), Jakarta, Indonesia, 2 Jakarta Provincial Health Office, Jakarta, Indonesia, 3 Faculty of Public Health, Universitas Indonesia, Depok, Indonesia, 4 World Health Organization, Geneva, Switzerland

☯ These authors contributed equally to this work.
* adrianna.bella@cisdi.org

## Abstract

### Background

Understanding the actual prevalence of COVID-19 transmission in the community is vital for strategic responses to the pandemic. This study aims to estimate the actual infection of COVID-19 through a seroprevalence survey and to predict infection fatality rate (IFR) in Tanjung Priok, the hardest-hit sub-district by the COVID-19 in Jakarta, Indonesia.

### Methods

We conducted a venous blood sampling (phlebotomy) to 3,196 individuals in Tanjung Priok between Nov 23, 2020, and Feb 19, 2021 to detect their antibodies against SARS-CoV-2. Using an enumerator-administered questionnaire, we collected data on the respondents' demographic characteristics, COVID-19 test history, COVID-19 symptoms in the last 14 days, comorbidities, and protective behaviours during the last month. We employed descriptive analysis to estimate the seroprevalence and IFR.

### Findings

The prevalence of Antibody against SARS-CoV-2 was 28.52% (95% CI 25.44–31.81%), with the result being higher in females than males (OR 1.20; 95% CI 1.02–1.42). By the end of the data collection (February 9, 2021), the cumulative cases of COVID-19 in Tanjung Priok were reported to be experienced by 9,861 people (2.4%). Those aged 45–65 were more likely to be seropositive than 15–19 years old (OR 1.42; 95% CI 1.05–1.92). Nearly one third (31%) of the subjects who developed at least one COVID-19 symptom in the last 14 days of the data collection were seropositive. The estimated IFR was 0.08% (95% CI 0.07–0.09), with a higher figure recorded in males (0.09; 95% CI 0.08–0.10) than females (0.07; 95% CI 0.06–0.08), and oldest age group (45–65) (0.21; 95% CI 0.18–0.23) than other younger groups.

**Data Availability Statement:** The data is owned by the Center for Indonesia's Strategic Development Initiatives (CISDI) and can be made available for scientific purposes upon request. A data sharing

agreement will be needed to ensure privacy and data handling. A request for deidentified data and a data dictionary can be requested to the corresponding author (adrianna.bella@cisdi.org) or secretariat@cisdi.org.

**Funding:** This study is fully funded by the Center for Indonesia's Strategic Development Initiatives (CISDI). We disclose that the institution is an independent non-governmental organization which does not have any conflict of interest with any potential external bodies. The funder had no role in study design, data collection, and data analysis.

**Competing interests:** The authors have declared that no competing interests exist.

## Conclusion

An under-reporting issue was found between the estimated COVID-19 seroprevalence and the reported cumulative cases in Tanjung Priok. More efforts are required to amplify epidemiological surveillance by the provincial and local governments.

## Introduction

Since the first case of COVID-19 in Indonesia detected on March 6, 2020, the country has been enormously affected by the ongoing pandemic [1]. After over a year of constant battle, Indonesia has confirmed 4,178,164 cases and 139,682 deaths due to COVID-19 as of September 16th, 2021 [1]. Among other cities, Jakarta, the capital city, is frequently recorded as the city with the highest cases in Indonesia [2]. As one of the main entry points for international and domestic travels in Indonesia, Jakarta is considerably a vulnerable hotspot for COVID-19 transmission [3]. Moreover, the severe overcrowding in Jakarta [4] has put further strain as it has been linked to the increase in COVID-19 cases in urban communities [5].

In an immediate response to the pandemic, the World Health Organization (WHO) has urged countries worldwide, including Indonesia, to optimize the strategies of comprehensive public health interventions [6]. However, limited resources and capacities [7] have created discrepancies in testing capacity across provinces in Indonesia [8]. In the early phase of the pandemic, most provinces in Indonesia were struggling to reach the minimum standard of 1:1000 population weekly testing as suggested by the WHO [9]. Some provinces (Jakarta, West Sumatera, and Yogyakarta) have reported current success in achieving the said standard [8]. However, the COVID-19 positivity rate, an epidemiological criterion to assess COVID-19 transmission [10], is never below 5% [8]. Even worse, as the current screening and testing priority are dedicated primarily to the symptomatic cases [11], the presence of asymptomatic cases [12] in Indonesia leads the high possibility of underestimation between the reported and actual figure of COVID-19 transmission [13]. It raises an urgency to conduct a seroprevalence study to estimate the actual prevalence of COVID-19 infections in the community. Not only is this approach pivotal to obtain the actual proportion of the people who have already had antibodies against the coronavirus, but given the demographic heterogeneity in Indonesia, having granular data on the community level may help the government provide better-targeted public health responses during this pandemic [14].

Until recently, extensive seroprevalence studies have been conducted on a population basis in many countries [15]. The results generally found a higher prevalence of the actual COVID-19 infection than the reported one [15], such as a study in Guilan, Iran that discovered a 22% seroprevalence, 323 times as high as the reported cases [16]. To date, only one study explores COVID-19 seroprevalence in Indonesia, particularly in East Java,—a province that frequently contributes largely to the high cumulative COVID-19 cases in the country [17]. Therefore, it is vital to conduct more seroprevalence studies, especially in the special capital region of Jakarta that constantly records the highest cases of COVID-19 in Indonesia [2].

This study aims to investigate the seroprevalence by population's age and sex in Tanjung Priok, a sub-district in Jakarta with the highest COVID-19 case, i.e., 2,983 confirmed cases by November 1, 2020 [18]. Additionally, we predict the infection fatality rate (IFR) in Tanjung Priok based on the calculated seroprevalence. This study is the second seroprevalence study in Indonesia and among the limited seroprevalence studies in the Southeast Asia region and the low- and middle-income countries. The results of this study may provide a way forward for

the policy makers to improve the surveillance and preventive measures of COVID-19 response in Tanjung Priok by providing the actual estimate of COVID-19 infection by demographic status and the predicted value of IFR.

## Methods

### Study design and participants

We recruited 3,196 individuals in Tanjung Priok sub-district, Jakarta from November 23, 2020, to February 19, 2021, to participate in this cross-sectional study. The inclusion criteria of the participants were: i) residing in Tanjung Priok sub-district for at least six months, ii) aged 15–65 years, iii) not in a COVID-19 isolation/quarantine period, iv) being able to come to the location of blood sample collection.

The minimum sample size was determined with the Lemeshow *et al.* [19] approach using a 3% margin of error, 30% estimation of COVID-19 prevalence, 95% confidence level, and a design effect of 2. The sample was selected through a cluster sampling in two stages: 1) random hamlet selection using the Probability Proportional to Size (PPS) sampling and 2) random household invitation and on-site respondent recruitment. A total of 42 random hamlets resulted from the PPS method with replacement. Several days prior to the data collection in each hamlet, households were randomly selected, and the members were invited to the location of blood sample collection. Anticipating low participation rate of the invited respondents, we recruited uninvited (on-site) respondents coming from the same hamlets as the invited ones at the blood collection sites. The overall percentages of invited respondents and on-site respondents were 61.86% and 38.14%, respectively.

Blood samples were drawn through venous blood sampling (phlebotomy) and run in Roche Elecsys Anti-SARS-CoV-2 (Specificity 99.81%; sensitivity 0–6 days, 7–13 days, and more than 14 days after a positive Rapid Test Polymerase chain reaction (RT-PCR) results were 65.5%, 88.1%, and 100%, respectively) diagnostic in Prodia laboratory. Through an enumerator-administered questionnaire, the respondents detailed their demographic characteristics, COVID-19 test history, COVID-19 symptoms in the last 14 days, comorbidities, and COVID-19 protective behaviours in the past month (see S1 Appendix for a complete list of the questions asked in Indonesian and English).

### Study variables

We included demographic characteristics, protective behaviours, and symptoms in the analysis. The sociodemographic characteristics include sex, age groups, education level, working status, and working conditions during the COVID-19 pandemic. Working status indicates the main activities during the last two weeks, categorized into 1) not in the labour force, 2) unemployed, 3) formal workers, and 4) informal workers. The formal and informal workers were identified under four working conditions: 1) working as usual, 2) working from home, 3) partly working from home, and 4) being temporarily laid off.

The protective behaviours are measured as the frequency of conducting behaviours to prevent COVID-19 infection within the last month, which include washing hands with soap/hand sanitizer, carrying a hand sanitizer outside of the house, wearing a mask outside of the house, limiting mobility, not shaking hands with others, and physical distancing. The mobility index was created using weighted factor analysis of questions related to travelling behaviours, namely 1) hanging out with more than two people; 2) going to public places; 3) attending places of worship; 4) using mass public transportation; 5) visiting supermarkets and 6) visiting health-care facilities. In terms of COVID-19 symptoms, we used a dummy variable of having at least

one of seven COVID-19 symptoms experienced within the last fourteen days, including fever, cough, sore throat, headache, vomiting, diarrhea, and myalgia [20].

## Statistical analysis

We employed a post-stratification weighting method to adjust our sample to match the distribution of sex and age in the population (as of 2020) of Tanjung Priok sub-district, Jakarta, Indonesia, obtained from Jakarta Open Data [18]. Using five age groups (15–19 years, 20–24 years, 25–34 years, 35–44, and 44–65 years), we determined the weight as the sex-age proportion in Tanjung Priok divided by sex-age proportion in our sample. Crude (unweighted) and adjusted (weighted) descriptive statistics were employed to estimate the seroprevalence and IFR. The IFR was calculated using a formula in Ioannidis [21] that divides the number of COVID-19 deaths by the estimated number of infected people. The estimated number of infected people was inferred from the seroprevalence result by multiplying the total population and the adjusted seroprevalence. All analyses were performed using STATA 15, and the results were considered significant at 5% significance level.

## Ethical consideration

The study was carried out following the guidelines of the Declaration of Helsinki on personal data handling. All respondents signed informed consent prior to the collection of blood samples and questionnaire completion. The study was approved by the Ethics Committee of Atma Jaya Catholic University (Number: 0889A/III/LPPM.PM.10.05/08/2020) in August 2020.

## Results

### Prevalence of anti-SARS-CoV-2 antibodies

As illustrated in Table 1, the adjusted prevalence of anti-SARS-CoV-2 in Tanjung Priok during the data collection period was 28.52% (95% CI 25.44–31.81%). The value of the weighted prevalence was quite similar to the unadjusted prevalence of 29.91% (95% CI 26.65–33.39%).

Regarding data on previous RT-PCR results as shown in Table 2, only 1.53% (49) of the total respondents reported a history of positive RT-PCR. Among those who previously had positive RT-PCR results, 80.80% had also positive serology tests, while around 19% of them did not show a discoverable antibody against SARS-CoV-2.

### Respondent characteristics

Table 3 shows the socio-demographic characteristics, protective behaviours, and COVID-19 symptoms of the sample based on the prevalence of anti-SARS-CoV-2 antibodies. The percentage of seropositive was significantly higher in females than males (30.42% vs. 26.64%; OR 1.20; p = 0.029) and among the 45–65 years than 15–19 years (25.19% vs. 32.36%; OR 1.42; p = 0.023). There was no observed difference of the seroprevalence among respondents with different education levels, working status, and working conditions.

**Table 1. Seroprevalence of antibody against SARS-CoV-2 in Tanjung Priok.**

|  | n | Crude seroprevalence | | Adjusted seroprevalence | |
|---|---|---|---|---|---|
|  |  | % | 95% CI | % | 95% CI |
| **All samples (n = 3,196)** |  |  |  |  |  |
| **Seropositive to SARS-CoV-2 Antibodies** | 956 | 29.91 | 26.65–33.39 | 28.52 | 25.44–31.81 |

**Table 2. Seroprevalence of antibody against SARS-CoV-2 by self-reported RT-PCR result.**

| | n | Crude seroprevalence | | Adjusted seroprevalence | |
|---|---|---|---|---|---|
| | | % | 95% CI | % | 95% CI |
| **Having a history of Positive RT-PCR Result* (n = 49)** | | | | | |
| **Seropositive to SARS CoV-2 Antibodies** | 39 | 79.59 | 63.65–89.67 | 80.80 | 63.57–91.03 |
| Seropositive with positive RT-PCR < 2 weeks before the serology test (n = 17) | 12 | 70.59 | 42.29–88.71 | 74.83 | 48.24–90.46 |
| Seropositive with positive RT-PCR 3–4 weeks before the serology test (n = 11) | 10 | 90.91 | 51.01–98.97 | 90.06 | 52.43–98.67 |
| Seropositive with positive RT-PCR 1–2 months before the serology test (n = 13) | 10 | 76.92 | 43.53–93.51 | 81.25 | 51.52–94.64 |
| Seropositive with positive RT-PCR > 2 months before the serology test (n = 8) | 7 | 87.50 | 28.30–99.20 | 80.14 | 18.43–98.63 |

*RT-PCR result before the serology test and self-reported by respondent.

In terms of protective behaviours to prevent COVID-19 infection, we found no significant differences in the seroprevalence across protective-behaviour variables. We also found that there were no significant differences in the seroprevalence between symptomatic and asymptomatic respondents. Among respondents who developed at least one symptom for the last 14 days before the serology test (21.15%), only around 31% had a seropositive result. The most reported COVID-19 symptoms were cough (8.79%), headache (8.32%), fever (4.10%), and myalgia (3.68%). A separate analysis of each COVID-19 symptom revealed that only fever (OR 1.86; p = 0.002), cough (OR 1.44; p = 0.012), and diarrhea (OR 1.97; p = 0.02) were associated with the higher seropositive estimates.

## Estimated IFR

As seen in Table 4, the estimated IFR calculated from the adjusted seroprevalence was 0.08 (95% CI 0.07–0.09), which means, approximately eight out of 10,000 people infected with COVID-19 in Tanjung Priok would die by Feb 19, 2021. The available data of COVID-19 deaths showed that the estimated IFR was slightly higher among male (0.09; 95% CI 0.08–0.10) than female (0.07; 95% CI 0.06–0.08). The inferred IFR by age groups indicated that IFR was estimated to increase with age, hence, the IFR of people aged 45–65 years (0.21; 95% CI 0.18–0.23) was more than twenty and five times as high as those aged 25–34 years (0.01; 95% CI 0.01–0.02) and 35–44 years (0.04; 95% CI 0.03–0.05), respectively. The age range of respondents in the seroprevalence study had limited our IFR estimates to 15–65 age groups. It is also worth noting that the calculation of the IFR included only the reported COVID-19 deaths from the confirmed cases, ignoring the figure of the probable deaths due to the unavailability of the data.

## Discussion

This study discovered that 28.52% (95% CI 25.44–31.81%) of people in Tanjung Priok sub-district had detectable antibodies against SARS-CoV-2 during the data collection period (November 23, 2020, to February 19, 2021). Approximately four out of five people (80.80%, 95% CI 63.57–91.03%) with a history of positive RT-PCR developed identifiable antibodies against SARS-CoV-2, while antibodies in one-fifths of the respondents might either have not developed or diminished at the time of blood sample collection.

**Table 3. Seroprevalence by demographic characteristic.**

| | Crude seroprevalence | | | | Adjusted seroprevalence | | | |
|---|---|---|---|---|---|---|---|---|
| | Positive | Negative | OR (95% CI) | p-value | Positive | Negative | OR (95% CI) | p-value |
| | N (%) | N (%) | | | N (%) | N (%) | | |
| **(a) Demographic Characteristics** | | | | | | | | |
| **Sex (N = 3,196)** | | | | | | | | |
| Male | 342(27.65%) | 895(72.35%) | Ref | | 427(26.64%) | 1,177(73.36%) | Ref | |
| Female | 614(31.34%) | 1,345(69.66%) | 1.19(1.02–1.40) | 0.026* | 484(30.42%) | 1,107(69.58.%) | 1.20(1.02–1.42) | 0.029** |
| **Age group, years (N = 3,196)** | | | | | | | | |
| 15–19 | 68(25.47%) | 199(74.53%) | Ref | | 84(25.19%) | 249(74.81%) | Ref | |
| 20–24 | 76(29.80%) | 179(70.20%) | 1.24 (0.85–1.83) | 0.268 | 100(29.68%) | 236(70.32%) | 1.25(0.85–1.86) | 0.259 |
| 25–34 | 116(25.66%) | 336(74.34%) | 1.01 (0.71–1.43) | 0.954 | 182(25.65%) | 527(74.35%) | 1.02(0.72–1.46) | 0.893 |
| 35–44 | 217(28.74%) | 538(71.26%) | 1.18 (0.86–1.62) | 0.306 | 222(27.19%) | 594(72.81%) | 1.11(0.80–1.54) | 0.535 |
| 45–65 | 479(32.65%) | 988(67.35%) | 1.42 (1.06–1.91) | 0.021* | 324(32.36%) | 677(67.64%) | 1.42(1.05–1.92) | 0.023* |
| **Education level (N = 3,196)** | | | | | | | | |
| No education/ primary school | 104(29.46%) | 249(70.54%) | Ref | | 73(26.80%) | 204(73.08%) | Ref | |
| Junior secondary school | 146(30.61%) | 331(69.39%) | 1.06(0.78–1.43) | 0.722 | 120(27.48%) | 316(72.52%) | 1.03(0.75–1.42) | 0.835 |
| Senior secondary school | 519(30.75%) | 1,169(69.25%) | 1.06(0.83–1.37) | 0.634 | 515(29.83%) | 1.212(70.17%) | 1.16(0.89–1.51) | 0.272 |
| University or higher | 187(27.58%) | 491(72.42%) | 0.91(0.69–1.21) | 0.525 | 203(26.77%) | 556(73.23%) | 1.00(0.74–1.35) | 0.990 |
| **Working status (N = 3,196)** | | | | | | | | |
| Not in the labour force | 481(30.83%) | 1,079(69.17%) | Ref | | 404(29.54%) | 966(70.46%) | Ref | |
| Unemployed | 56(27.59%) | 147(72.41%) | 0.85(0.62–1.18) | 0.345 | 69(25.86%) | 199(74.14%) | 0.83(0.59–1.17) | 0.294 |
| Formal worker | 265(31.18%) | 585(68.82%) | 1.02(0.85–1.22) | 0.862 | 289(29.60%) | 688(70.40%) | 1.00(0.82–1.22) | 0.974 |
| Informal worker | 154(26.42%) | 429(73.58%) | 0.81(0.65–1.00) | 0.046* | 148(25.52%) | 432(74.48%) | 0.82(0.65–1.03) | 0.089 |
| **Working condition during the pandemic (N = 1,433)** | | | | | | | | |
| Working as usual | 242(28.84%) | 597(71.16%) | Ref | | 93(27.15%) | 250(72.85%) | Ref | |
| Working from home | 101(27.08%) | 272(72.92%) | 1.37(0.93–2.03) | 0.115 | 59(30.49%) | 134(69.51%) | 1.18(0.77–1.80) | 0.451 |
| Partly working from home | 57(33.73%) | 112(66.27%) | 1.10(0.83–1.43) | 0.529 | 228(27.29%) | 608(72.71%) | 1.00(0.74–1.36) | 0.962 |
| Temporarily laid off | 19(36.54%) | 33(63.46%) | 1.55(0.84–2.85) | 0.158 | 22(36.57%) | 39(63.43%) | 1.55(0.80–2.99) | 0.194 |
| **(b) Protective behaviours** | | | | | | | | |
| **Washing hands with soap/hand sanitizer (N = 3,196)** | | | | | | | | |
| Otherwise | 117(32.14%) | 247(67.86%) | Ref | | 125(31.99%) | 265(68.01%) | Ref | |
| Always/often | 839(29.63%) | 1,993(70.37%) | 0.89(0.70–1.12) | 0.324 | 787(28.04%) | 2,019(71.96%) | 0.83(0.64–1.07) | 0.149 |
| **Bringing hand sanitizer outside of the house (N = 3,196)** | | | | | | | | |
| Otherwise | 818(29.86%) | 1,921(70.14%) | Ref | | 779(28.47%) | 1,957 (71.53%) | Ref | |
| Always/often | 138(30.20%) | 319(69.80%) | 1.02(0.82–1.26) | 0.886 | 133(28.85%) | 327 (71.15%) | 1.02(0.81–1.29) | 0.874 |
| **Using a mask outside of the house (N = 3,196)** | | | | | | | | |
| Otherwise | 185(31.79%) | 397(68.21%) | Ref | | 174(30.40%) | 398(69.60%) | Ref | |
| Always/often | 771(29.50%) | 1,843(70.50%) | 0.90(0.74–1.09) | 0.275 | 738(28.11%) | 1,886(71.89%) | 0.90(0.73–1.10) | 0.302 |
| **Mobility index (N = 3,196)** | | | | | | | | |
| Low | 473(29.40%) | 1,136(70.60%) | Ref | | 408(28.50%) | 1,023(71.50%) | Ref | |
| High | 483(30.43%) | 1,104(69.57%) | 1.05(0.90–1.22) | 0.522 | 504(28.54%) | 1,261(71.46%) | 1.00(0.85–1.18) | 0.981 |
| **Not shaking hands with others (N = 3,196)** | | | | | | | | |
| Otherwise | 61(33.89%) | 119(66.11%) | Ref | | 63(33.26%) | 126(66.74%) | Ref | |
| Always/often | 895(29.68%) | 2,121(70.32%) | 0.82(0.60–1.13) | 0.231 | 849(28.22%) | 2,159(71.78%) | 0.79(0.56–1.11) | 0.177 |
| **Maintaining physical distancing (N = 3,196)** | | | | | | | | |
| Otherwise | 350(31.47%) | 762(68.53%) | Ref | | 335(29.65%) | 795(70.35%) | Ref | |
| Always/often | 606(29.08%) | 1,478(70.92%) | 0.89(0.76–1.04) | 0.159 | 576(27.90%) | 1,489(72.10%) | 0.92(0.77–1.09) | 0.332 |
| **(c) COVID-19 Symptoms** | | | | | | | | |

*(Continued)*

**Table 3.** (Continued)

| | Crude seroprevalence | | | | Adjusted seroprevalence | | | |
|---|---|---|---|---|---|---|---|---|
| | Positive | Negative | OR (95% CI) | p-value | Positive | Negative | OR (95% CI) | p-value |
| | N (%) | N (%) | | | N (%) | N (%) | | |
| **Having at least one symptom (N = 3,196)** | | | | | | | | |
| No | 737(29.34%) | 1,775(70.66%) | Ref | | 701(27.83%) | 1,819(72.17%) | Ref | |
| Yes | 219(32.02%) | 465(67.98%) | 1.13(0.95–1.36) | 0.175 | 210(31.08%) | 466(68.92%) | 1.17(0.96–1.43) | 0.125 |
| **Fever (N = 3,196)** | | | | | | | | |
| No | 903(29.45%) | 2,163(70.55%) | Ref | | 851(27.95%) | 2,208(72.05%) | Ref | |
| Yes | 53(40.77%) | 77(59.23%) | 1.65(1.15–2.36) | 0.006* | 55(41.90%) | 76(58.10%) | 1.86(1.25–2.76) | 0.002* |
| **Cough (N = 3,196)** | | | | | | | | |
| No | 855(29.22%) | 2,071(70.78%) | Ref | | 811(27.83%) | 2,104(72.17%) | Ref | |
| Yes | 101(37.41%) | 169(62.59%) | 1.45(1.12–1.88) | 0.005* | 100(35.65%) | 181(64.35%) | 1.44(1.08–1.90) | 0.012* |
| **Sore throat (N = 3,196)** | | | | | | | | |
| No | 928(29.78%) | 2,188(70.22%) | Ref | | 879(28.30%) | 2,229(71.70%) | Ref | |
| Yes | 28(35.00%) | 52(65.00%) | 1.27(0.80–2.02) | 0.315 | 32(36.61%) | 55 (63.39%) | 1.46(0.88–2.43) | 0.147 |
| **Myalgia (N = 3,196)** | | | | | | | | |
| No | 921(30.10%) | 2,139(69.90%) | Ref | | 883(28.20%) | 2,196(71.33%) | Ref | |
| Yes | 35(25.74%) | 101(74.26%) | 0.80 (0.54–1.19) | 0.278 | 30(24.59%) | 89(75.41%) | 0.81(0.53–1.25) | 0.342 |
| **Headache (N = 3,196)** | | | | | | | | |
| No | 869(29.74%) | 2,053(70.26%) | Ref | | 832(28.41%) | 2,098(71.59%) | Ref | |
| Yes | 87(31.75%) | 187(68.25%) | 1.10(0.84–1.43) | 0.487 | 79(29.79%) | 187(70.21%) | 1.07(0.80–1.43) | 0.651 |
| **Vomiting (N = 3,196)** | | | | | | | | |
| No | 947(29.85%) | 2,226(70.15%) | Ref | | 901(28.41%) | 2,269(71.59%) | Ref | |
| Yes | 9(39.13%) | 14(60.87%) | 1.51(0.65–3.50) | 0.336 | 11(41.57%) | 15(58.43%) | 1.79(0.73–4.37) | 0.200 |
| **Diarrhea (N = 3,196)** | | | | | | | | |
| No | 931(29.68%) | 2,206(70.32%) | Ref | | 883(28.20%) | 2,248(71.80%) | Ref | |
| Yes | 25 (42.37%) | 34 (57.63%) | 1.7 (1.03–2.93) | 0.037* | 28(43.64%) | 37(56.36%) | 1.97(1.11–3.48) | 0.020* |

**Table 4. The estimated IFR.**

| Category | Population size* | Adjusted seroprevalence, % (95% CI) | Estimated number of infections (95% CI) | Confirmed COVID-19 death** | IFR, % (95% CI) |
|---|---|---|---|---|---|
| **Total population** | | | | | |
| Total | 419,555 | 28.52 (25.44–31.81) | 119,657 (106,735–133,460) | 97 | 0.08 (0.07–0.09) |
| **Sex** | | | | | |
| Male | 211,367 | 26.64 (23.60–29.92) | 53,308 (49,882–63,241) | 50 | 0.09 (0.08–0.10) |
| Female | 208,188 | 30.42 (26.63–34.50) | 63,331 (55,440–71,825) | 47 | 0.07 (0.06–0.08) |
| **Age** | | | | | |
| 15–19 | 30,955 | 25.19 (19.29–32.16) | 7,798 (5,971–9,955) | 0 | 0 |
| 20–24 | 31,330 | 29.68 (23.46–36.77) | 9,299 (7,350–11,520) | 0 | 0 |
| 25–34 | 65,945 | 25.65 (20.81–31.17) | 16,915 (13,723–20,555) | 2 | 0.01 (0.01–0.02) |
| 35–44 | 75,858 | 27.18 (22.52–32.43) | 20,618 (17,083–24,601) | 9 | 0.04 (0.03–0.05) |
| 45–65 | 93,046 | 32.36 (28.68–36.27) | 30,110 (26,686–33,748) | 62 | 0.21 (0.18–0.23) |

* Based on the total population in Tanjung Priok district in 2020 by Opendata Jakarta (2021).

** Confirmed COVID-19 death is the cumulative death of confirmed COVID-19 people on February 19th, 2021, from Jakarta public health office.

The finding of seroprevalence showed that the estimated number of infected individuals was around twelve times as high as the cumulative cases reported by February 19, 2021 (2.33%; obtained by dividing the number of cumulative cases in Tanjung Priok sub-district on February 19[th], 2021 (9,792) recorded by Jakarta public health office, with the total population in 2020 [18]), indicating a wide discrepancy (under-estimation) between the estimate and cumulative reported cases that might have resulted from two factors: 1) insufficient testing in the respective area and 2) asymptomatic nature of the COVID-19 infection [22]. Although the number of tests in the sub-district in February 2021 had fulfilled the WHO recommendation [9] (1.01 per 1,000 population per week; calculated by dividing total RT-PCR in Tanjung Priok community health centre on February 12–18, 2021 (425) with the total population in 2020 [18]), the 7-day-rolling-average test per new confirmed case in the same period (2.85; obtained by dividing total RT-PCR in Tanjung Priok community health centre on February 12–18, 2021 (425) with the total confirmed cases on February 12–18, 2021 (149) [18]) remained far below the WHO standard of 10–30 tests per confirmed case [23]. Curbing the community transmission may require even more rigorous, population-scale testing which has been demonstrated as an evidently effective strategy in some countries to reduce SARS-CoV-2 infection [24]. Indonesia's testing per case in the given period was far below that of some other countries like India (56.5), Malaysia (20), and the Philippines (18.2) [25]. The asymptomatic nature of COVID-19 estimated at around 1.4%-78.3% [26] has compounded the detection of the COVID-19 cases worldwide, including Indonesia. The problem of asymptomatic cases in Indonesia is worsened by the scarcity of RT-PCR test kits, resulting in test prioritization for people with COVID-19 symptoms [12].

We found that the seroprevalence estimate in this study was quite similar to that of serological surveys in Northern France (25.9%) [27] and Qatar (30.4%) [28]. Other findings ranged from 0.6% to 59% [29], implying a heterogeneity of seroprevalence across geographical areas [21] that could largely be mediated by several underlying factors, such as different transmission rates due to population density, testing capacity and characteristics, the time frame of data collection, the state of the epidemic, and the policy regulations implemented in the respective location of the study [30]. However, these serological studies shared an ability to identify the gap between the cumulative reported cases and the seroprevalence estimate. A major under-ascertainment (twelve times) in the present study was corroborated by a meta-analysis by Bobrovitz *et al.*, [29] reporting a gap nearly 6.7 times to 602.5 times between the estimated seroprevalence and the cumulative reported incidence across studies.

The 1.2-time as high proportion of women as men with seropositivity (30.42% vs 26.64%) was in line with the national COVID-19 database during the period of data collection [2]. However, the result of this study differs from a meta-analysis of 968 seroprevalence studies that did not find any significant difference in seroprevalence between females and males [29]. To date, the proportion pattern of the infected population by sex remains vague because of a high variation in sex-disaggregated data reported by 168 countries [31].

Some respondents in the present study (21.15%) reported to have developed at least one symptom of COVID-19 in the past 14 days, one-third of them being seropositive. The wide share of asymptomatic subjects between those with antibodies against COVID-19 in this study was similar to a systematic review by Oran and Topol [32] reporting 50–60% asymptomatic cases in 16 clinical studies. The finding of this study, however, should be interpreted with caution as the onset of COVID-19 infection was not identified regardless of the symptoms developed within the last 14 days of the recall period. While differences of seroprevalence between symptomatic and asymptomatic respondents were absent in the present study, previous findings showed otherwise [33, 34]. This could be best explained by the inability of most seroprevalence studies [22] in identifying the period of SARS-CoV-2 infection among the subjects.

Our study demonstrated no significant differences in seropositive status between people who complied with COVID-19 protective behaviours and those who did not. This is, however, by no means indicating that performing protective behaviours, such as washing hands with soap/using sanitizer, wearing a mask when outside, and maintaining physical distance provide no protection from COVID-19 transmission. As COVID-19-related protective behaviours tend to alter over time due to societal and environmental factors [35, 36], this study was unable to draw the actual association of protective behaviours and the seropositive status. This finding is consistent with Roederer *et al.* [37] who found no evidence of any correlation between the presence of SARS-CoV-2 antibodies and people's adherence to COVID-19 preventive measures.

Our analysis of seroprevalence by age group shows that a higher risk of seropositivity was evident in those aged 45–65 years than 15–19 years. While this finding was consistent with a study in Hungary discovering a higher seropositive rate of 40–64 age group [38], it was different from the national and provincial data showing higher COVID-19 prevalence among younger age groups (19–45 and 19–48 years old, respectively) [2, 39]. Contrarily, WHO asserted no differences in COVID-19 infection across age groups from the currently available knowledge [40]; therefore, the variation found among studies and reports might be due to people's mobility and the age structure of the population [41]. Furthermore, it is worth mentioning that in our study as well as the Hungarian study [38], the oldest age group largely comprises the age structure as well as seropositive results, hence, leading to the possibility of significant results in older adults.

Additionally, we estimated the IFR to be 0.08% by estimating the proportion of deaths among the estimated number of actual infections. This result, however, should be treated cautiously considering the underreported data of COVID-19-related deaths [42] that may stem from 1) the uncounted deaths from probable cases and 2) the significant gap (3.3 times) between the number of COVID-19 burial procedures and the reported COVID-19 deaths in Jakarta [42]. The IFR in our result was lower than the global average (0.15%) and some advanced countries like Canada (0.59%) and England (1.16%) [20]. Considering the less-developed reporting system and healthcare facilities in Indonesia compared to those in advanced countries [43], we proposed that the actual IFR in Tanjung Priok was potentially far higher than our estimation.

The present study could provide a more robust analysis if not for limitations and challenges. First, the lack of information on the types of antibodies against SARS-CoV-2 (IgG and IgM) made it difficult to distinguish the current and past infections among the subjects. Also, the challenging process to obtain randomised participants made us mix the invited (randomised) and on-site (non-randomised) respondents as our subjects. We maintained the sample representativeness by applying the age-sex post-stratification weight. Another source of weakness in this study is the relatively long data collection period of almost three months, which might suffer from pandemic dynamics due to potentially increased mobility during an end-of-year holiday. As the data were limited to the self-reported data, there were potential social desirability bias, information bias, and recall bias regarding COVID-19 history, protective behaviours, and COVID-19 symptoms. Regarding IFR, the inadequate data reporting system of COVID-19 deaths in the respective provincial government is one uncontrolled factor that affects the calculation of IFR.

Notwithstanding the limitations, the study has provided a better understanding of the estimated percentage of people infected by SARS-CoV-2 in Tanjung Priok sub-district. To the best of our knowledge, this study is the first seroprevalence study conducted in a sub-district in Jakarta, Indonesia's capital city that is frequently recorded as having the highest cumulative COVID-19 cases in the country [2]. Our study contributes significantly to the currently scarce

seroprevalence studies in Indonesia as well as in the Southeast Asia region and the low- and middle-income countries.

## Conclusion

This study aims to estimate the actual SARS-CoV-2 infection in Tanjung Priok sub-district, Jakarta between November 23, 2020, and February 19, 2021. Engaging 3,196 participants, we estimate that approximately three out of ten people (28.52%, 95% CI 25.44–31.81%) in Tanjung Priok developed detectable antibodies against SARS-CoV-2. A serious under-ascertainment of COVID-19 cases was observed as the seroprevalence estimate reveals a twelvefold reported cumulative case. Being subjected to the available data of COVID-19 attributable deaths, we also calculate an estimated IFR of 0.08% (95% CI 0.07–0.09).

Concerning the limitation of this study, more robust studies on seroprevalence are vital to determine the estimated proportion of infected people in larger areas, such as cities, districts, or provinces in Indonesia, especially those with potentially high infection rate. Further seroprevalence research can explore different types of antibodies against SARS-CoV-2 to include the analysis of the infection period.

The findings of this paper have several implications for future policy practice. First, considering the under-ascertainment of reported cases, greater efforts are needed to further amplify not only epidemiological surveillance (testing and tracing measures) by the provincial government of Jakarta, but also the capacity of the health care at the primary health care centres and hospitals in Jakarta as an anticipation for the surge. Second, the potentially underestimated IFR also suggests further improvement of the death reporting system and the definition of COVID-19 death on national and subnational levels.

## Supporting information

**S1 Appendix. Survey questionnaire in Indonesian and English.**
(DOCX)

**S1 File. Research summary, challenges, and prospects.**
(DOCX)

## Author Contributions

**Conceptualization:** Olivia Herlinda, Adrianna Bella.

**Data curation:** Dimitri Swasthika Nurshadrina, Mochamad Thoriq Akbar.

**Formal analysis:** Dimitri Swasthika Nurshadrina, Mochamad Thoriq Akbar.

**Investigation:** Mochamad Thoriq Akbar, Sofwatun Nida.

**Methodology:** Adrianna Bella, Mochamad Thoriq Akbar, Iwan Ariawan.

**Project administration:** Adrianna Bella, Mochamad Thoriq Akbar, Sofwatun Nida.

**Resources:** Sofwatun Nida, Ngabila Salama.

**Supervision:** Olivia Herlinda.

**Visualization:** Dimitri Swasthika Nurshadrina.

**Writing – original draft:** Olivia Herlinda, Adrianna Bella, Gita Kusnadi, Dimitri Swasthika Nurshadrina.

**Writing – review & editing:** Olivia Herlinda, Adrianna Bella, Gita Kusnadi, Ngabila Salama, Iwan Ariawan, Diah Saminarsih.

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
