## [Decision Letter · Decision Letter 0]

9 Nov 2021

PONE-D-21-30389Seroprevalence of Antibodies against SARS-Cov-2 in the High Impacted Sub-district in Jakarta, IndonesiaPLOS ONE

Dear Dr. Bella,

Thank you for submitting your manuscript to PLOS ONE. After careful consideration, we feel that it has merit but does not fully meet PLOS ONE’s publication criteria as it currently stands. Therefore, we invite you to submit a revised version of the manuscript that addresses the points raised during the review process.

We look forward to receiving your revised manuscript.

Kind regards,

Sanjay Kumar Singh Patel, Ph.D.

Academic Editor

PLOS ONE

Journal Requirements:

2. If your study included minors, state whether you obtained consent from parents or guardians. If the need for consent was waived by the ethics committee, please include this information.

“This study is fully funded by the Center for Indonesia's Strategic Development Initiatives (CISDI), where most authors work as researchers, as one of its research projects. We disclose that the institution is an independent think tank organization which does not have any conflict of interest with any potential external bodies.”

Reviewers' comments:

Reviewer's Responses to Questions

**Comments to the Author**

1. Is the manuscript technically sound, and do the data support the conclusions?

Reviewer #1: Yes

Reviewer #2: Yes

2. Has the statistical analysis been performed appropriately and rigorously? 

Reviewer #1: Yes

Reviewer #2: Yes

3. Have the authors made all data underlying the findings in their manuscript fully available?

Reviewer #1: No

Reviewer #2: Yes

4. Is the manuscript presented in an intelligible fashion and written in standard English?

Reviewer #1: Yes

Reviewer #2: Yes

5. Review Comments to the Author

Reviewer #1: This investigation uses serology testing to detect the antibodies against SARS-CoV-2 in the population of Tanjung Priok in Jakarta, Indonesia to estimate the seroprevalence and infection fatality rate (IFR) in the community. The results show that the prevalence of antibody against SARS-CoV-2 was 28.52%. The estimated IFR was 0.08%. The big difference between the estimated COVID-19 seroprevalence and the reported cumulative cases in Tanjung Priok shows significant under-reporting of COVID-19 cases in this community. As this study presented some interesting information on the estimated percentage of people infected by SARS-CoV-2 in Tanjung Priok sub-district of Jakarta, it may be published yet not in priority. This is because, in the current stage of the pandemic, after several waves of SARS-COV-2 virus breakout worldwide, understanding the actual prevalence of COVID-19 transmission in the community might not be strategically important in fighting the COVID-19 pandemic when most of the COVID-19 cases are asymptomatic or mild. Contrary to the authors suggestion of putting more efforts to amplify epidemiological surveillance by the provincial and local governments, more efforts and resources should be reserved for the care and treatment of the severe cases of COVID-19 patients to save lives.

Reviewer #2: In the current research article entitled " Seroprevalence of Antibodies against 1 SARS-Cov-2 in the High Impacted Sub-district in Jakarta, Indonesia", by Herlinda et al., have studied/surveyed estimate the seroprevalence and infection fatality rate (IFR) in Tanjung Priok, the hardest-hit sub-district by the COVID-19 in Jakarta, Indonesia. Authors conducted venous blood sampling to 3,196 individuals in Tanjung Priok between Nov 23, 2020, and Feb 19, 2021. They found that, under-reporting is an issue between the estimated COVID-19 seroprevalence and the reported cumulative cases in Tanjung Priok. This article addresses a research topic of great interest, which is under intense investigation in the past 2 years and the manuscript is generally well-written. However, this reviewer has certain suggestions that would help produce a more comprehensive overview of the topic:

Suggestions:

1. The authors may additionally provide one Figure as summary, challenges, or prospect of the present study.

2. The authors should cross-check all abbreviations in the manuscript. Initially, define in full name followed by abbreviation.

3. The English of manuscript can be polished (minor).

---

## [Author Response · Author response to Decision Letter 0]

6 Dec 2021

Responses to editor's comments:

Dear Editor,

We thank you for the comments given to our manuscript which improved the quality of our paper significantly. On your suggestion to upload our lab protocol, unfortunately we did not do any lab procedures so there is no lab protocol available. 

Being specific on the other points you raised:

#1. We revised the manuscript style and file naming accordingly. 

#2. We included minors aged 15-17 years old and had obtained the consent through their guardians. 

#3. We have uploaded the questionnaire (in Indonesian and English) we used in this study into the system as the supplementary information (S1_Appendix). We have also added a new heading titled “Supporting information” in the manuscript.

#4. We have added the following statement "This study is fully funded by the Center for Indonesia's Strategic Development Initiatives (CISDI). We disclose that the institution is an independent non-governmental organization which does not have any conflict of interest with any potential external bodies. The funder had no role in study design, data collection, and data analysis."

#5. We have rechecked all the references and certain everything is in correct order and complete.

Response to Reviewer #1:

Dear reviewer #1, we thank you for the general appreciation of our work, and specific comments given that help to improve our manuscript. We believe it is important to understand the scale of this pandemic through seroprevalence survey as part of evaluation process of government policies and interventions. Ultimately, whether the testing and tracing capacity have been sufficient, which then may provide estimate how much more it should be improved. As well, though it is still in the central of debate, seroprevalence can be the basis for the government to pursue how much more vaccination rates should be moved forward, as well as in which demographics. Related to your further comment, we agree that by knowing the scale of the infection, the government can calculate and predict how much more care and treatment should be prepared for the coming wave, which we have incorporated in line 327-329 in the file named “Manuscript”.

Response to Reviewer #2:

Dear Reviewer #2,

We thank you for your generous comments. To address some of your specific points: 

#1. We have provided the Figure as a supplementary file (S2_File). We have also added a new heading titled “Supporting information” in the manuscript. 

#2. We have rechecked it again and make sure that 1) all abbreviations are mentioned when the spelled-out versions first appear in the manuscript and 2) all abbreviations are uniform throughout the manuscript, thank you for the kind reminder.

#3. Thank you, we have rechecked the accuracy of language that we used in the manuscript. Changes were tracked in the file “Revised Article with Changes Highlighted”.

---

## [Editor Report · Decision Letter 1]

14 Dec 2021

Seroprevalence of Antibodies against SARS-Cov-2 in the High Impacted Sub-district in Jakarta, Indonesia

PONE-D-21-30389R1

Dear Dr. Bella,

We’re pleased to inform you that your manuscript has been judged scientifically suitable for publication and will be formally accepted for publication once it meets all outstanding technical requirements.

Kind regards,

Sanjay Kumar Singh Patel, Ph.D.

Academic Editor

PLOS ONE

---

## [Editor Report · Acceptance letter]

16 Dec 2021

PONE-D-21-30389R1 

Seroprevalence of Antibodies against SARS-Cov-2 in the High Impacted Sub-district in Jakarta, Indonesia 

Dear Dr. Bella:

I'm pleased to inform you that your manuscript has been deemed suitable for publication in PLOS ONE. Congratulations! Your manuscript is now with our production department. 

Kind regards, 

on behalf of

Dr. Sanjay Kumar Singh Patel 

Academic Editor

PLOS ONE